# Enhancing Mezcal Production Efficiency by Adding an Inoculant of Native *Saccharomyces cerevisiae* to a Standardized Fermentation Must

**DOI:** 10.3390/foods14030341

**Published:** 2025-01-21

**Authors:** Armando H. Holguín-Loya, Adriana E. Salazar-Herrera, Nicolas O. Soto-Cruz, Manuel R. Kirchmayr, Christian A. Lopes, Juan A. Rojas-Contreras, Jesús B. Páez-Lerma

**Affiliations:** 1TecNM/Instituto Tecnológico de Durango, Blvd. Felipe Pescador 1830 Ote, Durango 34080, Durango, Mexico; 20040539@itdurango.edu.mx (A.H.H.-L.); 22041571@itdurango.edu.mx (A.E.S.-H.); jrojas@itdurango.edu.mx (J.A.R.-C.); 2Biotecnología Industrial, Centro de Investigación y Asistencia en Tecnología y Diseño del Estado de Jalisco, A.C. Camino Arenero 1227, El Bajío del Arenal, Zapopan 45019, Jalisco, Mexico; mkirchmayr@ciatej.mx; 3PROBIEN, Instituto de Investigación y Desarrollo en Ingeniería de Procesos, Biotecnología y Energías Alternativas, Universidad Nacional del Comahue, Neuquén 8300, Argentina; christianariellopes@gmail.com

**Keywords:** agave must fermentation, mezcal, inoculant implantation, volatile compounds

## Abstract

All traditional mezcal producers use artisan methods to produce mezcal. The low technological development in the elaboration processes results in low yield and high residual sugar concentration. First, this work optimized the concentration of initial sugars and yeast-assimilable nitrogen (YAN) in *Agave durangensis* juice fermentation at the laboratory level. A yield near 0.49 g EtOH/g sugar and a productivity of 1.54 g EtOH/L*h was obtained with an initial sugar concentration of 120 g/L and a YAN concentration of 0.227 g/L. Only *Saccharomyces cerevisiae* was found after 24 h of incubation at laboratory level, using MALDI-TOF identification. *Agave durangensis* heads crushed by the artisan process were used to test the inoculant performance. A mezcal yield of 11.6 kg agave/L of mezcal was obtained using the *S cerevisiae* inoculant and nitrogen addition, which was significantly different (*p* < 0.05) from other treatments. The population dynamics during fermentation were analyzed through isolation and identification using MALDI-TOF. Several yeast species (*Pichia kluyveri*, *Torulaspora delbrueckii*, *Zygosaccharomyces bailii*, and *Saccharomyces cerevisiae*) were found at the beginning of fermentation. Nonetheless, only *S. cerevisiae* was found at the end of fermentation. The implantation of the inoculant used was confirmed through the comparative analysis of amplification patterns of the GTG5 microsatellite of the strains identified as *S. cerevisiae*, finding that the inoculated strain proportion was greater than 80% of the yeast population. A technological alternative to increase the efficiency of the process is combining the addition of YAN and the inoculation of the native *S. cerevisiae*, which was isolated from artisan alcoholic fermentation of agave to produce mezcal.

## 1. Introduction

Mezcal is a beverage of Mexican origin, and its name comes from the Nahual “mexcalli”, which means “cooked agave”. It is defined as a distilled drink obtained from fermentation through spontaneous or cultured microorganisms of the juices of cooked agaves harvested within the Denomination of Origin [1,2]. This drink is made mainly through artisanal processes developed by small producers, using traditional techniques and tools inherited by the so-called “mezcal masters”. The core of the production procedure is fermentation carried out using cultured or uncultured yeasts, which in artisan production leads to slow fermentation, high concentrations of residual sugars, low concentrations of ethanol, and an increase in acidity derived from lactic acid bacteria that proliferate because of these problems [3,4,5,6,7].

There is a trend to modernize production methods [8,9], and one proposed change is adding an inoculant to improve the predictability and control of fermentation. *Saccharomyces cerevisiae* (*S. cerevisiae*) is a yeast frequently used in the alcoholic beverages industry to reduce the risk of incomplete fermentations and enhance the product’s stability, safety, and sensory quality [10,11,12]. Still, as reported previously [13], commercial strains can show poor growth because of the agave juice’s complexity, which contains high sugar concentrations, Maillard compounds, saponins, and furfural. Instead, selected native microorganisms are adapted to the conditions of the agave juice and can dominate fermentation [4].

The inoculation of a selected microorganism does not ensure the completion of the fermentation because it must adapt and compete with the native strains present. An inoculant is used so that the select strain dominates fermentation, obtaining maximum use of fermentable sugars, high product yield, and adequate organoleptic characteristics [14]. Since inoculation does not guarantee the implantation of the microorganism [15], it is necessary to monitor the evolution of the microbial population. Some techniques, such as microsatellite analysis, proteomic analysis, and matrix-assisted desorption ionization/desorption time-of-flight (MALDI-TOF) strain identification, help determine the inoculant implantation.

MALDI-TOF is an identification technique that has proven to be advantageous due to its effectiveness and speed since the organisms can be identified in a matter of minutes, regardless of their type, whether they are yeasts or bacteria, avoiding the steps before their differentiation, through minimum sample preparation [16,17]. Then again, intra-specific differentiation techniques, which involve distinguishing between different strains or isolates within a single species using microsatellites such as GTG5, have helped describe population dynamics and identify dominant native strains [18]. Other studies have shown that this technique provides good discrimination to study the diversity of prokaryotes with a genetic fingerprint that can be applied to microbial ecology and evolution studies [19].

On the other hand, a study carried out in cider production [20] highlighted the importance of the concentration of elements such as nitrogen, magnesium, and zinc, which are necessary for the optimal development of yeasts during fermentation. Nitrogen is a crucial element for yeast growth and metabolism. At the same time, magnesium and zinc are essential for various enzymatic reactions in yeast cells. The carbon/nitrogen ratio (C/N) relates to the mass of carbon and organic nitrogen available for microbial growth and metabolism, influencing bioprocesses development [21,22]. Low C/N values can mean excess available nitrogen, leading to growth inhibition or the generation of toxic compounds like ammonia. Contrariwise, high values show a lack of nitrogen and excess of the carbon source [22]. Different works [23,24,25] have described the effect of the C/N ratio, but other works [26,27,28] have reported that each strain has an affinity for a particular nitrogen source and that native strains are adapted to growth with nitrogen limitations.

This work aimed to evaluate the implantation of an inoculant of a selected native strain of *S. cerevisiae* during agave fermentation to produce mezcal. The better sugar and yeast-assimilable nitrogen (YAN) concentration was also determined to affect ethanol yield, fermentation productivity, and mezcal yield (kg agave per liter of mezcal).

## 2. Materials and Methods

### 2.1. Microbiological Material and Chemicals

The strain *Saccharomyces cerevisiae* ITD-00185, isolated from the alcoholic fermentation of agave musts [29], was used. This strain was previously characterized during agave juice fermentations [30]. It was highlighted for its good fermentative capacity (high sugar consumption, high ethanol production, and low acetic acid production), high ethanol tolerance, and neutral killer phenotype. The strain also showed a good capacity to produce volatile compounds, such as esters, alcohols, and terpenes.

Sulfuric acid (CAS: 7664-93-9), boric acid (CAS: 10043-35-3), hydrochloric acid (CAS: 7647-01-0), Disodium Phosphate (CAS: 7558-79-4), Monopotassium Phosphate (CAS: 7778-77-0), dichloromethane (CAS: 75-09-2), anhydrous sodium sulfate (7757-82-6), ammonium phosphate monobasic (CAS: 7722-76-1), and ethanol (CAS: 64-17-5) were supplied by Fermont (Monterrey, N.L., Mexico). Sodium hydroxide (CAS: 1310-73-2), fructose (CAS: 57-48-7), and glucose (CAS: 50-99-7) were purchased from Meyer^®^ (Tláhuac, CDMX, Mexico). Sodium chloride (CAS: 7647-14-5), glycine (CAS: 56-40-6), agarose (CAS: 9012-36-6), and methyl red (CAS: 493-52-7) were supplied by Fisher Scientific S.L. (Alcobendas, Madrid, Spain). Ninhydrin (CAS: 485-47-2) and Trizma base (CAS: 77-86-1) were supplied by Sigma-Aldrich^®^ (St. Louis, MO, USA). EDTA (CAS: 6381-92-6), and Potato Dextrose Agar (PDA) was supplied by CTR scientific^®^ (Monterrey, N.L. Mexico). Methylene blue (CAS: 7220-79-3) was supplied by Hycel (Zapopan, Mexico).

### 2.2. Raw Material

Cooked *Agave durangensis* was obtained from a mezcal factory in Durango, México. This factory uses the traditional cooking method described by Arellano-Plaza et al. [31]. In brief, after cutting the leaves, agave is placed underground in conical pit ovens containing stone and heated by burning wood. The stove is tapped with jute sacks and soil, maintaining this condition for 3–4 days. Five cooked agave samples were used to determine the sugar content in the cooked agave, and each sample was processed as follows. The first extract (concentrated juice) was obtained by directly pressing one kilogram of cooked agave. One liter of water at 60 °C was added to the pressed agave, and after 15 min of rest, it was pressed again to obtain a second extract. The same was repeated to obtain a third extract. Glucose and fructose were measured by triplicate to each extract using HPLC [32]. Glucose and fructose concentrations were summed to obtain the sugar concentration of the first (S1), second (S2), and third extracts (S3). Considering the extract volumes (vi), the sugar content of the agave was obtained using Equation (1). The presented result of this characterization was an average of the samples obtained.(1)$$Sugar\;content\;\left(\frac{g}{kg}\right)=\left[S_{1}\right]\left(v_{1}\right)+\left[S_{2}\right]\left(v_{2}\right)+\left[S_{3}\right]\left(v_{3}\right)$$

Total nitrogen was quantified by the micro-Kjeldahl method [33]. A sample of 0.1 g of dried juice was placed in a 100 mL Kjendahl flask with 0.03 g of a Selenium reagent mixture and 2 mL of sulfuric acid. It was heated to boil for 3 h until clarified. After cooling, 25 mL of distilled water and 10 mL of sodium hydroxide (40%) were added. This mixture was distilled and recovered in a beaker containing 10 mL of boric acid (4%) and mixed until 30 mL were obtained. Finally, the product was titrated by adding hydrochloric acid (0.02 N) until the solution turned green to purple. The ninhydrin method determined the YAN content [34]. Three test tubes were prepared by separately adding 2 mL of water for the Blank, 2 mL of the diluted sample (1:100), and 2 mL of glycine solution (2 mg/L) for the standard. One milliliter of ninhydrin color solution was added to each reaction tube. Afterward, the tubes were mixed, capped, and brought to a boiling water bath for 16 min. Then, they were cooled at 20 °C for 20 min, and 5 mL of dilution solution was added and mixed. Lastly, the optical density was read in a UV-VIS DR6000 HACH spectrophotometer (Loveland, CO, USA) at 570 nm, and the following equation calculated the YAN content:(2)$$YAN\;\left(\frac{\mathrm{mg}}{L}\right)=\frac{A_{S}-A_{B}-A_{C}}{A_{G}-A_{B}}\;(2)(d)$$
where $A_{S}$ is the sample absorbance, $A_{G}$ is the absorbance of the glycine standard, $A_{B}$ is the Blank absorbance, $A_{C}$ is the absorbance of the correction for darkness, d is the dilution factor, and 2 is the concentration of the glycine standard solution (mg/L).

### 2.3. Effects of Sugar and YAN Concentrations

The effect of sugar and YAN concentrations was evaluated employing a central composite design (CDC) using the Desing Expert^®^ software version 13.0.1.0 64 bit (Table 1).

CDC was conducted using fermentations inoculated with the selected yeast *S. cerevisiae* ITD-00185. Experiments were developed in triplicate using Erlenmeyer flasks (500 mL) at an operating volume of 300 mL of agave juice. Fermentation kinetics started with a cell density of 1 × 10^6^ cells/mL. Concentrated juice was obtained by directly pressing the cooked agave. Distilled water was added to adjust to the desired sugar concentration, adding ammonium phosphate as a source of YAN (Table 1). Fermentations were incubated for 48 h at 28 °C without stirring. Samples were taken every three hours to quantify biomass, glucose, fructose, and ethanol. Cell density was estimated using the standard curve of cell density (g/L) versus optical density (absorbance units) shown in Appendix A. From these results, the maximum specific growth rate ($\mu_{max}$) was obtained by modeling growth data using the integrated form of the logistic equation (Equation (3)) [35].(3)$$X(t)=\frac{X_{max}}{1+\left(\frac{X_{max}}{X_{0}}-1\right){e^{-}}^{\mu_{max}t}}$$
where t is the time (h), X is the biomass concentration (g/L), $X_{max}$ is the maximum biomass concentration (g/L), $X_{0}$ is the biomass concentration at t = 0 (g/L), and $\mu_{max}$ is the specific growth rate (h^−1^). Ethanol yield (*Y_EtOH/S_*) was calculated by dividing the ethanol produced by the sugars consumed at 48 h. Productivity (*Q_P_*) was obtained as the quotient of the concentration of ethanol produced at 48 h between the elapsed time.

### 2.4. Agave Fermentations at Pilot Plant Level and Implantation of Inoculant

In experimental condition C1, the must was supplemented with YAN (0.227 g/L) and inoculated with the selected strain. In experimental condition C2, it was augmented with YAN (0.227 g/L) but not inoculated. Finally, in experimental condition C3, it was neither supplemented with YAN nor inoculated.

Based on the working range determined by the central composite design, fermentations were carried out at the pilot plant level using stainless steel vats (20 L) at an operating volume of 10 L. Fermentations were performed at 28 °C for 48 h at an initial cell density of 1 × 10^6^ cells/mL. Biomass, glucose, fructose, and ethanol were quantified from samples taken every three hours for the first nine hours and then every six hours.

The population dynamics were studied using samples taken at 0, 24, and 48 h of fermentation. The samples were diluted in sterile distilled water up to 1 × 10^−6^. Once diluted, 200 µL were seeded on agar plates containing commercial medium Potato Dextrose Agar (PDA) of BD Bioxon™ from CTR scientific^®^ (Monterrey, N.L. Mexico). They were incubated at 28 °C for 48 h. After incubation, the colonies were selected randomly using Harrison’s disk technique, taking a statistically representative sample of 10% of the colonies from each plate. Adequate dilution was chosen when contenting 150 to 300 colonies [36].

The taxonomic identification of the isolated strains was made using MALDI-TOF. Biomass was directly spread in 96-well polished steel plates. Each sample was treated with 1.0 μL of 70% formic acid and allowed to dry; subsequently, 1.0 μL of the HCCA matrix solution was added (α-cyano-4-hydroxycinnamic acid) to a 10 mg/mL solution, which contained acetonitrile, water, and trifluoroacetic acid in a ratio of 50:47.5:2.5, respectively. The treated plates were processed using a Microflex LT mass spectrometer and Flex Control software (version 3.0, Bruker Daltonics GmbH, Bremen, Germany), generating mass spectra of an approximate size between 2 and 20 kDa. The output files were compared with the BDAL v.8 database generated by Gallegos et al. [37].

Implantation of the selected strain was determined by comparing the specific banding pattern of the isolates identified as *S. cerevisiae*. PCR amplification was conducted through the GTG5 microsatellite, using 12.5 µL GoTaq^®^ G2 Colorless Master mix of Promega (Madison, WI, USA) and 7.5 µL of water. Nuclease-free, 2 µL of 30% GTG5 primer (5′-GTGGGTGGTGGTGGTG-3′), and 3 µL of DNA sample were used. The amplification conditions were denaturation at 94 °C for 4 min, 30 cycles at 95 °C for 30, 45° for 1 min, and 65 °C for 8 min, ending with a final extension at 65 °C for 16 min [20]. Amplified samples were separated on a 2% (*w*/*v*) agarose gel made with 0.5× TAE buffer (0.04 M Tris-acetate, 1.0 mM EDTA, pH 8.0) by electrophoresis under the following conditions: initial migration 5 min at 110 V and subsequently 90 min at 80 V. The gel was stained with ethidium bromide for visualization under UV light. A 1 Kb molecular weight marker was used for band size comparison [19].

After fermentation, the must was distilled twice in a stainless steel distiller with an operation volume of 5 L to obtain mezcal. The cooling liquid was fed to the condenser at −8 °C, obtaining the distillate in fractions of 100 mL. The alcoholic degree of each fraction was determined using a densimeter as described by the Mexican standard NMX-V-013-NORMEX-2013 (2014) [38]. Distillation was stopped when it reached a fraction with 5%Alc. Vol. The fractions were combined, and a second distillation (rectification) was carried out until a fraction with 10% Alc. Vol. was obtained. The fractions obtained were mixed cold for subsequent analysis of this mezcal.

Minor compounds in the mezcals were analyzed as follows: liquid–liquid extraction mixed with 15 mL mezcal (30% ethanol), 15 mL dichloromethane, and 4.5 g NaCl. Subsequently, the organic phase was recovered and dried with sodium anhydrous sulfate. It was concentrated in a Kuderna-Danish column up to 1.5 mL. The samples were analyzed in duplicate by gas chromatography coupled to mass spectrometry (GC/MS) [39].

### 2.5. Analytical Techniques

Biomass quantification was made by optical density (OD) at 600 nm using a UV-VIS DR6000 HACH spectrophotometer (Loveland, CO, USA) in a 1:10 dilution. Cell density was estimated using the standard curve of cell density (g/L) versus optical density (absorbance units) shown in Appendix A.

Glucose, fructose, and ethanol concentrations were obtained by HPLC Agilent Technologies 1200 (Santa Clara, CA, USA) on an Aminex^®^ (Dublin, Ireland) HPX-87H+ column (300 mm × 7.8 mm) under the following conditions: column temperature 65 °C, refractive index detector temperature 60 °C, injection volume 1 µL, and mobile phase sulfuric acid 0.5 mM at a constant flow rate of 0.6 mL/min [40].

Minor compound analysis was performed by GC/MS using an Agilent 7890A gas chromatograph equipped with an Agilent 5975C mass spectrometric detector (Santa Clara, CA, USA) following that reported by Acosta-García et al. [39]. In brief, the samples (1 µL) were injected in split mode (70:1) in an HP-FFAP column (30 m length × 0.32 mm inner diameter × 0.25 μm thickness; Agilent Technologies) to separate the compounds. High-purity helium (99.999%) was used as a carrier gas at a 1 mL/min constant flow. The injector temperature was 180 °C. The oven temperature was set to 40 °C for 3 min, then increased to 52 °C at 3 °C/min for 1 min, then increased to 200 °C at 10 °C/min, and held at 200 °C for 15 min. On the other hand, the mass spectrometer was operated at 230 °C, with an ionization voltage of −70 eV and SCAN mode (1.6 scans per second). An alkane ladder was injected to calculate the Kovats retention index of each compound. AMDIS software (v. 2.73, build 149.31) was used to identify the compounds.

### 2.6. Statistical Analysis

For the analysis of the results of the fermentative parameters, a one-way analysis of variance was used, with a post hoc method by Tukey’s test at *p* ≤ 0.05, using Minitab 19^®^ software. On the other hand, Design Expert 13^®^ statistical software was used for the response surface analysis of product, performance, and productivity parameters.

## 3. Results and Discussion

### 3.1. Fermentations at Laboratory Level

The concentrated agave juice, which was used as a base to formulate the different CDC experiments, had a sugar content of 301.99 ± 15.89 g/L and a total nitrogen content of 0.378 ± 0.001 g/L, of which 0.084 ± 0.003 g/L was YAN.

Figure 1 shows the results obtained from CDC to evaluate the effect of sugars and YAN concentrations on fermentation parameters such as specific growth rate (*μ_max_*), ethanol production, productivity (*Q_P_*), and yield product/substrate (*Y_EtOH/S_*). The complete results used to build the response surfaces shown in Figure 1 are in Appendix A.

The parameter $\mu_{max}$ showed an inverse relationship with the initial sugar concentration but a direct relationship with the YAN content of the medium (Figure 1A). Authors like De León et al. [41] reported similar results at concentrations greater than 150 g/L in alcoholic fermentations using a minimal medium with the yeast extract. Nitrogen is, after carbon, the most critical element for developing yeasts and the correct evolution of fermentation [21]. It was also reported that the content of assimilable nitrogen in agave is low, mainly due to its degradation in the cooking process through Maillard reactions [26].

Figure 1B depicts that the more significant formation of ethanol was directly related to the initial amount of substrate and YAN. Similar results were obtained for productivity (*Q_P_*), finding values near 1.54 g EtOH/Lh when concentrations of sugar and YAN were high (Figure 1C). On the other hand, the YAN content benefited the yield of ethanol concerning sugar consumption (*Y_EtOH/S_*) but not sugar concentration (Figure 1D). An approximate 140 mg/L of YAN is necessary to carry out adequate alcoholic fermentation, depending on the concentration of sugars, the agave must used, and the type of yeast used. Still, an excess in the presence of nitrogen can cause inhibitory effects [22,25,33]. Stoichiometrically, the maximum theoretical *Y_EtOH/S_* is 0.51 g of ethanol produced per gram of glucose consumed [41]. Then, the *Y_EtOH/S_* values presented here are between 78 and 96% of the maximum theoretical, equivalent to about 0.49 g of ethanol produced per gram of glucose consumed. It presents a challenge in initial sugar determination, as sugars diffuse into the juice, complicating the process of yield determination [42]. These results demonstrate that adequate yeast growth and good ethanol production were obtained at sugar concentrations close to 120 g/L and a high YAN content (0.227 g/L). Therefore, these are the conditions to which the agave juice will be adjusted for experiments at the pilot plant level.

Samples taken at 0, 24, and 48 h were used to monitor population dynamics at the laboratory level. The fermentations started inoculating the selected strain of *S. cerevisiae*. Still, no treatment was applied to eliminate the native population contained in the cooked agave. Then, at the beginning of fermentation, a greater abundance of strains of *S. cerevisiae* and bacteria of the genera *Gluconobacter*, *Bacillus*, and *Acinetobacter* were found (Appendix A). However, *S. cerevisiae* notably advanced in displacing other microorganisms at 24 h. After 48 h of fermentation, the dominance of *S. cerevisiae* was evident since it was practically the only species present (Figure 2).

This behavior indicates that *S. cerevisiae* progressively takes control and conclusively dominates fermentation. Larralde et al. [13] reported that some strains, mainly commercial ones, have poor growth in agave fermentations due to their complexity. Agave juice contains various inhibitory compounds, such as Millard compounds and furfural, and high concentrations of sugars and saponins. *Agave durangensis* is one of the agaves with the highest saponin content [43]. Then, the native strain *S. cerevisiae* ITD 00185 exhibits a high ability to adapt to local conditions and dominate fermentation.

### 3.2. Fermentation at the Plant Pilot Level

The sugar content of the cooked *Agave durangensis* was 238.63 ± 32.38 g/kg. The variability of this result depends on various factors, such as the plant’s maturity, environmental conditions, soil quality, production region, water availability, among others [44]. The sugar content of agave was reported to be 20–32%. Species below 20% are considered low quality, while those with 25–30% are considered good quality [45,46]. The fermentation juice was prepared by calculating the amount of agave needed to obtain a sugar concentration of 120 g/L after adding 10 L of water. The results for the C1, C2, and C3 experimental conditions at the pilot plant scale are shown in Table 2.

Table 2 shows a relative consumption of 82.1% only in experimental condition C1 (sugar concentration of 120 g/L, YAN concentration of 0.227 g/L, and inoculating). The inoculant, a key component in starting fermentation, was crucial in this condition. Fermentation conditions C2 and C3 showed an increase in the concentration of sugars due to the diffusion of these into the medium, as reported previously [40]. It means that, after 48 h, sugar consumption did not begin in these experimental conditions. The fermentations were extended up to 120 h to obtain mezcal in both situations. Then, the inoculant allowed a rapid start of fermentation. Bisson [46] attributed the speed of sugar consumption mainly to the amount of biomass present and its fermentative capacity. Additionally, various authors highlighted the need to adjust the C/N ratio in traditional fermentations to avoid fermentation stops, inhibition by substrate, formation of unwanted compounds, increase volatile compound production, among other characteristics benefiting fermentation [22,24,47,48,49].

Traditional agave fermentation was described as a variable process due to the artisan methods used, which can be slow with a duration between 8 and 30 days or even stopped. Low yields and residual sugar concentrations of up to 60 g/L are commonly obtained [3,50,51]. After 120 h of fermentation, the sugar consumption for conditions C2 and C3 was 94.77% and 34.86%, respectively (Appendix A). The effect of adding the inoculant and YAN was evident when comparing the volumes of mezcal produced with the fermentation time. In the case of the inoculated fermentation to which YAN was added (C1), the fermentation lasted 48 h, and 0.36 L of mezcal were obtained. It is equivalent to 7.5 × 10^−3^ L of mezcal per hour of fermentation. On the contrary, when YAN was added without inoculant (C2), the fermentation lasted 120 h, and 0.3 L of mezcal was produced, equivalent to 2.5 × 10^−3^ L of mezcal for each hour of fermentation. Thus, a process three-fold more efficient was observed when using the inoculant.

On the other hand, traditional fermentation (without YAN and inoculant, C3) lasted 120 h, and 0.1 L of mezcal was distilled (0.83 × 10^−3^ L of mezcal for each hour of fermentation). The sole addition of nitrogen increased the process’s efficiency by threefold. When comparing the traditional fermentation (C3) to the one carried out with the addition of YAN and the inoculant (C1), the latter is ninefold more efficient. Under condition C1, the volume of mezcal produced yielded 11.61 kg agave per liter of mezcal (Table 2). In comparison, condition C2 yielded 13.93 kg agave per liter of mezcal (Appendix A). This substantial increase in efficiency and the shorter fermentation time underscore the crucial role of inoculation and nitrogen supplementation in enhancing mezcal production. Arellano et al. [31] reported yields between 12 and 18 kg agave per liter of mezcal when using *Agave durangensis*, further solidifying this understanding.

Minor compound analysis of mezcals demonstrates a high diversity of compounds (Figure 3), such as aldehydes, alcohols, acids, ketones, and esters. The mezcal from traditional fermentation (C3) had mainly aldehydes, alcohols, and ketones. The addition of YAN (C2) promoted the formation of esters and acetic acid. The addition of YAN and the inoculant (C1) promoted the formation of many alcohols and some esters without acetic acid production. The traditional producers from Durango expose cooked agave to the open air for two or three days before adding the water to start fermentation [41,52]. It probably promotes the proliferation of bacteria [52] and yeasts, allowing a greater diversity of compounds in conditions C2 and C3 compared to condition C3. Then, the presence of the inoculant could limit the native microbiota development, leading to a different minor compound mixture. The samples analyzed are divided into two branches, grouping conditions C2 and C3, of which there were differences mainly in the presence of acid compounds; both samples present compounds with sweet notes (ethyl octanoate, 1-butanol, 2-methyl-1-propanol) [53] and floral notes (phenyl ethyl alcohol) [54]. However, it is necessary to conduct an in-depth study, quantifying the minor compounds and performing a sensory analysis. This analysis can determine whether the amount of these compounds correlates with consumer perception, potentially influencing their preferences and purchasing decisions.

The strains isolated during fermentation at the pilot plant level were analyzed to determine the population dynamics. The results show the presence of native yeasts (Figure 4A) of *S. cerevisiae*, *Z. bailii*, *T. delbruekii*, and *P. kluyveri* as main yeasts in spontaneous fermentation. Their variety changed as the fermentation progressed since *P. kluyveri* and *T. delbruekii* were eliminated. Meanwhile, *S. cerevisiae* remained the most abundant yeast at 24 and 48 h, followed by *Z. bailii*. It is the more recurrent behavior during the alcoholic fermentation of agave, where non-*Saccharomyces* strains can grow well in the initial stages of the process, with *S. cerevisiae* ending the fermentation [13]. Nonetheless, in some cases, yeasts such as *K. marxianus*, *S. diversa*, *T. delbrueckii*, *Z. bailii*, and *P. kluyveri* can be found at the end of fermentation, although in smaller proportions [40]. In the case of inoculated fermentation (Figure 4B), only strains of *S. cerevisiae* were identified in the three times analyzed.

Since *S. cerevisiae* predominates in inoculated and spontaneous fermentations, it was crucial to determine if the selected strain established itself and dominated fermentation. Notably, a proportion of the total isolates of *S. cerevisiae* in non-inoculated conditions coincides with the selected strain (Figure 5A). This strain was isolated from a mezcal factory in El Mezquital [29], located more than 100 km from the factory that provided the agave for the present study. It may indicate that the strain has a wider distribution than expected in Durango. However, further studies are required to verify this.

Figure 5B shows that the selected *S. cerevisiae* ITD-00185 progressively eliminated native *S. cerevisiae* strains and predominated at the end of the inoculated fermentation. Gil-Diaz et al. [55] and Lange et al. [56] described similar results during studies of the implantation of inoculants in wine fermentation, reporting that the inoculated microorganism guided the process. The decrease in fermentation time and the mezcal yields obtained can be directly related to the fermentative capacity of the selected strain (*S. cerevisiae* ITD-00185) and the addition of YAN.

## 4. Conclusions

The laboratory results demonstrated that adequate yeast growth and good ethanol production were obtained at sugar concentrations close to 120 g/L and a high YAN content (0.227 g/L). Therefore, these were the conditions to which the agave juice was adjusted for experiments at the pilot plant level. It was also demonstrated that *S. cerevisiae* progressively took control and conclusively dominated fermentation.

At the pilot plant level, the present work demonstrated a significant increase in efficiency and a shorter fermentation time, underscoring the significant role of inoculation and nitrogen supplementation in enhancing mezcal production. Moreover, the present research demonstrated the successful implantation of the selected strain, *S. cerevisiae* ITD-00185. This successful implantation paves the way for practical applications of our research in the field, instilling confidence in its potential.

## Figures and Tables

**Figure 1 foods-14-00341-f001:**
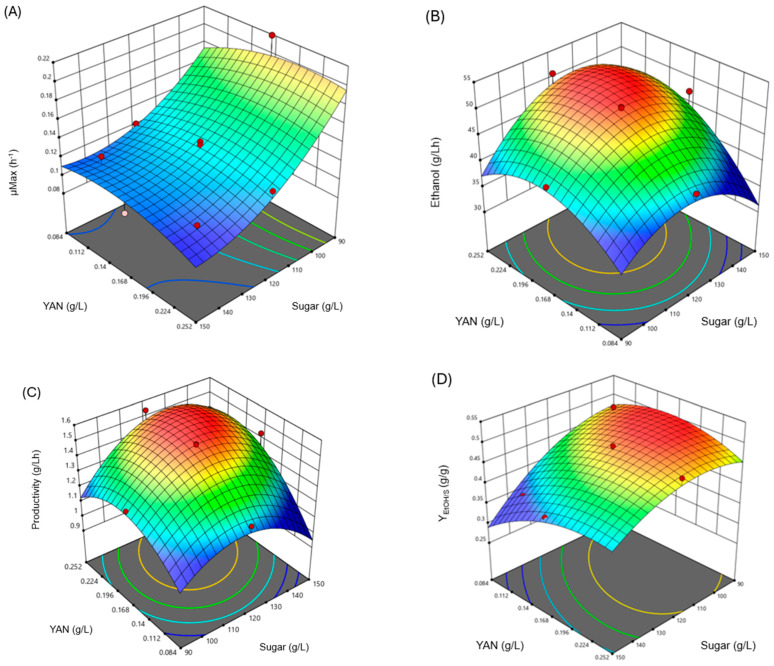
Effect of sugars and YAN on (**A**) maximum specific growth rate (μ_max_), (**B**) ethanol production (*EtOH*), (**C**) yield (*Y_EtOH/S_*), and (**D**) productivity (*Q_p_*) at laboratory level.

**Figure 2 foods-14-00341-f002:**
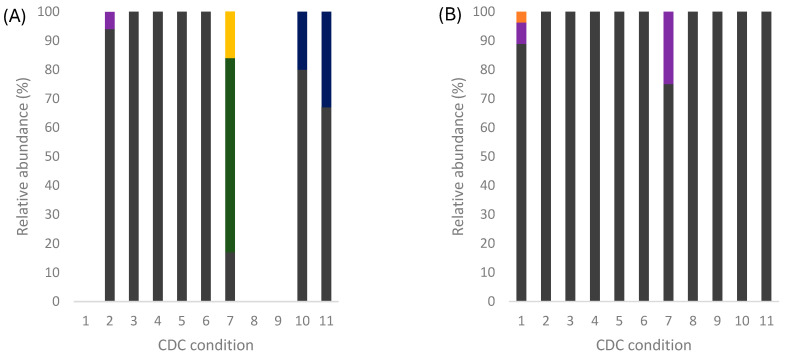
Relative abundance of strains identified by MALDI-TOF from CDC fermentations in agave juice at 0 (**A**) and 48 h of fermentation (**B**). ■
*Bacillos subtilis*, ■
*Brevibacillus centrosporus*, ■
*Acinetobacter radioresistens*, ■
*Acetobacter cerevisiae*, ■
*Gluconobacter oxydans*, and ■
*Saccharomyces cerevisiae*.

**Figure 3 foods-14-00341-f003:**
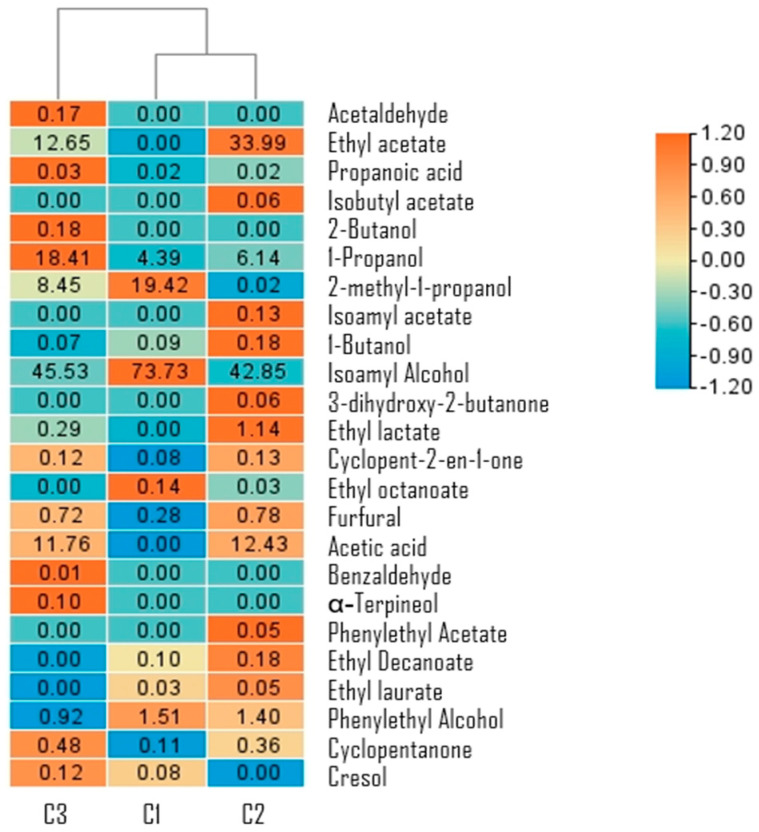
Relative abundance of volatile compounds determined for L-L extraction followed by concentration and GC/MS analysis of mezcals. Traditional fermentation (C3), fermentation with inoculum (C2), and fermentation with YAN and inoculum (C1).

**Figure 4 foods-14-00341-f004:**
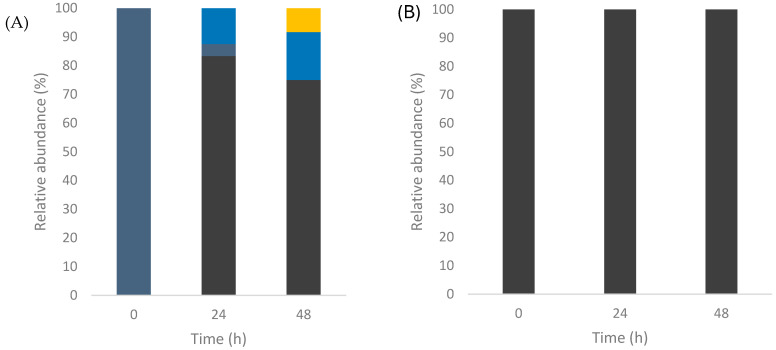
Relative abundance of strains identified by MALDI-TOF in spontaneous fermentation of agave at pilot plant level (**A**) and inoculated fermentation added with YAN (120 g/L and 0.227 g/L) at pilot plant level (**B**). ■
*T. delbrueckii*, ■
*Z. bailii*, ■
*P. kluyveri*, ■
*S. cerevisiae*.

**Figure 5 foods-14-00341-f005:**
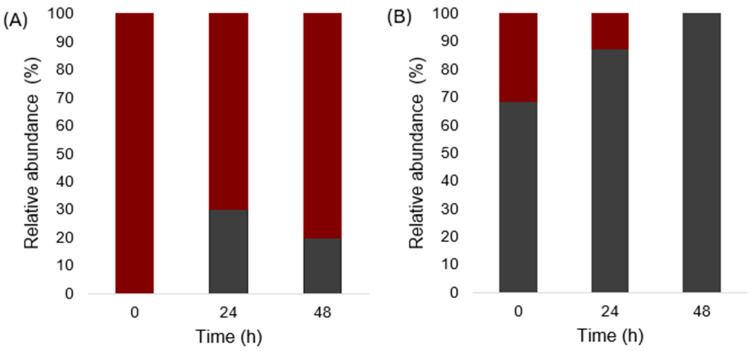
Relative abundance of *S. cerevisiae* strains analyzed by GTG5 for spontaneous fermentation at pilot plant level (**A**) and inoculated fermentation added with YAN (120 g/L and 0.227 g/L) at pilot plant level (**B**). ■
*S. cerevisiae* (*native*), ■
*S. cerevisiae* (*ITD-00185*).

**Table 1 foods-14-00341-t001:** Central composite design (CDC) experiments for evaluating effect of sugar content (glucose + fructose) and yeast-assimilable nitrogen (YAN) on alcoholic fermentation of agave juice.

Experiment	Sugar (g/L)	YAN (g/L)
1	90	0.168
2	120	0.084
3	98.7	0.108
4	141.2	0.108
5	120	0.168
6	150	0.168
7	120	0.252
8	141.2	0.227
9	120	0.168
10	98.7	0.227
11	120	0.168

**Table 2 foods-14-00341-t002:** Determination of agave fermentation parameters at pilot plant level. *S*_0_ (sugar concentration at beginning of fermentation), *S*_f_ (sugar content at 48 h), sugar consumption (percentage of sugar consumption at 48 h), mezcal (volume obtained from fermentations performed during 48 h), Y_kg agave/L mezcal_ (yield in kilograms of agave used to produce one liter of mezcal).

No.	Conditions			*S*_0_ (g/L)	*S*_f_ (g/L)	Sugar Consumption (%)	Mezcal (L)	Y_kg agave/L mezcal_
	Sugar (g/L)	YAN (g/L)	Inoculum (Cells/mL)					
C1	120	0.227	1 × 10^6^	118.67 ± 7.94	21.23 ± 3.65	82.10 ± 3.07	0.36	11.61
C2	120	0.227	-	110.85 ± 5.52	141.24 ± 0.26	-	-	-
C3	120	-	-	113.53 ± 5.64	136.20 ± 5.17	-	-	-

## Data Availability

The original contributions presented in the study are included in the article/Appendix A, further inquiries can be directed to the corresponding author.

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
