# Peer review of "Enhancing Mezcal Production Efficiency by Adding an Inoculant of Native Saccharomyces cerevisiae to a Standardized Fermentation Must"

_foods, 2025, doi:10.3390/foods14030341_

Round 1

Reviewer 1 Report

Comments and Suggestions for Authors

The manuscript titled “Enhancing The Mezcal Production Efficiency by Adding an Inoculant of Native Saccharomyces cerevisiae to a Standardized Fermentation must” (Manuscript ID: foods-3373183) optimized the initial sugar and yeast-assimilable nitrogen (YAN) concentrations during the fermentation of Agave durangensis juice at the laboratory level. After an initial sugar concentration of 120 g/L and a YAN concentration of 0.227 g/L were achieved, S. cerevisiae progressively took control and conclusively dominated the fermentation. As a result, significant increases in efficiency and shorter fermentation time were noted at the pilot plant level. It is an interesting work for practice, but many questions exist in the present version of this manuscript. Thus, authors have to make a major modification to their manuscript before resubmitting to the journal Foods. Specific points are issued as follows.

1/Line 47-48, Saccharomyces cerevisiae should be written in italic. 

2/Line 86, the abbreviation should be noted when “Saccharomyces cerevisiae firstly appeared.

3/Line 102, the meanings of S1/S2/S3and v1/v2/v3 in Equation 1 should be indicted clearly. .

4/Usually, microorganisms including bacteria and yeast often are counted in Colony-Forming Units (CFU). “cells/mL” is Ok for the enumeration of yeast?

5/Line 118-119, “From these results, the specific growth rate (μmax), as well as the ethanol yield (YEtOH⁄S), and productivity (QP) were calculated”. The methods for the calculations are what? Not clear.

6/Line 123-124, “with an inoculum of 1×106 cells/mL”. “Inoculum” indicates what? Should be cleared.

7/Line 128, producer information on the medium “Potato Dextrose Agar (PDA)” should be added.

8/Line 137, producer information for Microflex LT instrument should be indicated, including Instrument model, manufacturers, the city where the manufacturer is located. Please check the whole text.

9/Line 140, “The analyses were compared with the BDAL v.8 database”. The web sit of “BDAL v.8 database” should be indicated clearly.

10/Line 166, the meanings of Xmax, X0, t in Equation 2 should be cleared.

11/In the section Materials and Methods, five different agave samples were collected (see line 97), but why the results of the agave samples only from one group were presented?

12/Line 181-183, “Fig. 1 shows the results obtained from the CDC to evaluate the effect of sugars and YAN concentrations on fermentation parameters”. Which fermentation parameters” were affected? Should be indicated clearly.

13/Line 224 and line 293, “S. cerevisiae” should be written in italic. Please check the whole text.

14/For figure 1, the fonts are too small to read clearly, and the fonts for the axes in the sub-figures should be enlarged to facilitate reading.

15/For figure 2, the CDC condition of abscissa axis refer what? Not clear.

16/Line 237-241, should move the descriptions to the section Materials and Methods or delete.

17/Line 244-245, “Ykg pine/L mezcal (Yield in kilograms of pine used to produce one liter of Mezcal)”, pine is right? In the section Introduction, mezcal is obtained from cooked agave, and thus what the role of pine plays? (see line 269-275).

18/Line 264-268, 1) The amounts of mezcal obtained were 0.360 L in 48 h of fermentation (C1) and 0.3 L in 120 h of fermentation (C2), and it means that the addition of YAN allowed a 3-fold higher mezcal production than in traditional must (C3), demonstrating the significant impact of nitrogen supplementation on the fermentation process. it is right? 2) What is the relationship between line 264-266 and line 266-268?

19/Line 141-151, no corresponding results were presented in the section Results and Discussion.

20/For figure 3, 1) how the “spontaneous fermentation” group treated? 2) In sub-figure A, yellow means what? Should be cleared.

21/For the section References, formats should be unified.

22/For the section Abstract, line 17-18, “A yield near 0.49 g EtOH/g sugar and a productivity of 1.54 g EtOH/L*h” are not mentioned in the text.

Author Response

Thank you for your comments. A file with the corrections is attached.

Reviewer 2 Report

Comments and Suggestions for Authors

In this paper, the authors mainly studied the effect of the addition of saccharomyces cerevisiae and yeast-assimilable nitrogen on the production efficiency of mezcal. There are some innovations in the manuscript, but there are many problems in the pages of the manuscript, so it is recommended to overhaul. Some problems are listed as follows:

Major problem:

1.      The author mentioned five different agave samples in the selection of raw materials, but did not mention these five samples in the follow-up experiment, and the author needs to be clear about this.

2.      Why does the author analyze ethanol (etoh) and ethanol concerning (yetoh/s) separately? In terms of results, these two compounds are same, and the author needs to give the reasons.

3.      In 3.1, why did the author choose CDC 2,4,5,6,7,10,11 groups to study microbial populations instead of all groups of cdc?

4.      In 3.2, it is suggested that the author supplement the experiment of inoculating S. cerevisiae without adding nitrogen to explore the effect of nitrogen on the fermentation process.

Minor issues:

1.      In this paper, the volatile compounds of mezcal were not determined, but volatile compounds were mentioned in the keywords, which the author should modified.

2.      The author's introduction of ethanol production (EtOH) and maximum specific growth rate (𝜇𝑚𝑎𝑥) in FIG. 1 does not correspond to the order of the figures. It is suggested that the author clearly distinguish the meanings of figures A, B, C and D.

3.      In 3.2, The author mentions that "It is important to highlight that an appreciable proportion of S. cerevisiae in noninoculated conditions coincides with the selected strain ", does not correspond to Figure 4, the author needs to correct.

Author Response

(The authors gave the same response as above.)

Reviewer 3 Report

Comments and Suggestions for Authors

1. What are the criteria for selecting yeast strains? Is it solely based on a good fermentative capacity (characterized by high sugar consumption, high ethanol production, and low acetic acid production)? While this indeed helps address current fermentation issues, it may also lead to significant changes in product flavor due to the introduction of the strain. Under natural fermentation conditions, what is the abundance of this strain? What other flavor compounds does this strain produce during fermentation?

2. The experimental design is overly simplistic, resembling more of a process optimization effort rather than an in-depth scientific study, and thus it is not qualified for publication in the high-quality journal Foods.

3. The quality of fermented beverages is closely related to flavor compounds. This paper only studied the impact of inoculum size on sugar consumption and ethanol production. It should include an analysis of flavor compounds and a comparative analysis of the flavors between inoculated and non-inoculated fermentation samples.

4. Basic sensory analysis of inoculated and non-inoculated fermentation samples should be provided.

Comments on the Quality of English Language

The English could be improved to more clearly express the research.

Author Response

(The authors gave the same response as above.)

Reviewer 4 Report

Comments and Suggestions for Authors

Dear Authors, I have read the MS entitled Enhancing the mezcal production efficiency by adding an inoculant of native Saccharomyces cerevisiae to a standardized fermentation must. 

This study optimized the fermentation of Agave durangensis juice for mezcal production by improving sugar and yeast-assimilable nitrogen (YAN) levels. 

I have some observations:

-please check journal requirements and decide on wether to add a subchapter regarding used chemicals and equipment and their origin

-please expand on the type pf vegetal material used - how were the 5 samples different?  (line 97) what conditions were used for cooking the agave?

-please add a short description of the used methods of analysis, unless the journal has strict recommendations against it (line 103-104)

-please explain why distilation was stopped at 5, respectivelly 10 % vol (line 156)

-please give some context regarding environmental conditions, as it can influence the concentrations of fermentable compounds in the raw matter

-please explain in the beggining what Bagasse is and why it is important for this study

-was there any type of sensorial analysis used? please specify

Comments on the Quality of English Language

-please correct English all over the text, there are many small mistakes.

-please use italics when mentioning the names of microorganisms

-line 133 - please rephrase sentence as it does not make sense

Author Response

Thank you for your comments, a file with the corrections is attached.

Round 2

Reviewer 1 Report

Comments and Suggestions for Authors

This manuscript titled “Enhancing The Mezcal Production Efficiency by Adding an Inoculant of Native Saccharomyces cerevisiae to a Standardized Fermentation Must” (No. foods-3373183R2) maximized the concentration of starting sugars and yeast-assimilable nitrogen (YAN) in the fermentation of agave durangensis juice, which provides a notable improvement in efficiency and a shortened fermentation period, highlighting the important role that nitrogen supplementation and inoculation have in boosting mezcal production. Additionally, authors have revised their manuscript according to reviewers’ comments. No more points will be addressed.

Author Response

Dear Reviewer,

Response 1. We greatly appreciate your comments and suggestions, which have been very valuable in improving the analysis of our results and, ultimately, our manuscript. We have not made changes to the manuscript.

Reviewer 2 Report

Comments and Suggestions for Authors

In this paper, the authors mainly studied the effect of the addition of saccharomyces cerevisiae and yeast-assimilable nitrogen on the production efficiency of mezcal. There are some innovations in the manuscript, but there are still many problems in the pages of the manuscript, so it is recommended to minor revision. Some problems are listed as follows:

1.      In 3.1, why did the author choose CDC groups 1,8, and 9 with no relative abundance for microbial populations? Authors need to give reasons.

2.      In 3.2, it is suggested that the author supplement the experiment of inoculating S. cerevisiae without adding nitrogen to explore the effect of nitrogen on the fermentation process.

3.      In 3.2, There are errors in the interpretation of the temperament results by the authors, such as the addition of YAN and inoculant (C1) did not promote the production of many esters, which need to be corrected by the authors.

Author Response

Dear Reviewer,

We greatly appreciate your comments and suggestions, which have been very valuable in improving the analysis of our results and, ultimately, our manuscript.

COMMENTS 1: In 3.1, why did the author choose CDC groups 1, 8, and 9 with no relative abundance for microbial populations? Authors need to give reasons.
RESPONSE 1. The relative abundances of experimental conditions 1, 8, and 9 at 24 h were not obtained because excessive plate growth impeded isolating colonies for identification. Nonetheless, the results at 48 h are complete. In the corrected version, we included conditions 1, 8, and 9 without results to maintain consistency since these conditions have results at 48 h. We want to reiterate, as was answered in the first review, that this does not change the clear tendency observed since, at 24 h, S. cerevisiae made a notable advance in displacing other microorganisms, and, at 48 h of incubation, S. cerevisiae is practically the only species present. Respectfully, we do not consider that changes to the manuscript are needed.

COMMENTS 2: In 3.2, it is suggested that the author supplement the experiment of inoculating S. cerevisiae without adding nitrogen to explore the effect of nitrogen on the fermentation process.
RESPONSE 2: The effect of nitrogen addition is evident from the results reported in the manuscript. Consequently, we do not consider further experimentation necessary and respectfully request the reviewer to consider the corrected version of this part of the manuscript. As stated in the corrected version of the manuscript (Line 342), the sole addition of nitrogen increased the process's efficiency by threefold. Therefore, we do not consider that additional changes to the manuscript are needed.

COMMENTS 3: In 3.2, There are errors in the interpretation of the temperament results by the authors, such as the addition of YAN and inoculant (C1) did not promote the production of many esters, which need to be corrected by the authors.
RESPONSE 3. Thank you for pointing this out. We agree with this comment. Therefore, the sentence “The addition of YAN and inoculant (C1) promoted the formation of some alcohols and many esters without acetic acid production.” Changed to “The addition of YAN and inoculant (C1) promoted the formation of many alcohols and some esters without acetic acid production.” It has been highlighted in yellow in the manuscript [Line 354-355].